# Control with Patterns: A D-learning Method

**Quan Quan, Kai-Yuan Cai, and Chenyu Wang**
School of Automation Science and Electrical Engineering, Beihang University
(BUAA, Beijing University of Aeronautics and Astronautics), Beijing, China
{ qq_buaa, kycai, wangcy2023}@buaa.edu.cn

**Abstract:** Learning-based control policies are widely used in various tasks in the field of robotics and control. However, formal (Lyapunov) stability guarantees for learning-based controllers with nonlinear dynamical systems are difficult to obtain. We propose a novel control approach, namely Control with Patterns (CWP), to address the stability issue over data sets corresponding to nonlinear dynamical systems. For such data sets, we introduce a new definition, namely exponential attraction on data sets, to describe the nonlinear dynamical systems under consideration. The problem of exponential attraction on data sets is transformed into a problem of pattern classification one based on the data sets and parameterized Lyapunov functions. Furthermore, D-learning is proposed as a method to perform CWP without knowledge of the system dynamics. Finally, the effectiveness of CWP based on D-learning is demonstrated through simulations and real flight experiments. In these experiments, the position of the multicopter is stabilized using real-time images as feedback, which can be considered as an Image-Based Visual Servoing (IBVS) problem.

**Keywords:** Lyapunov Methods, Reinforcement Learning, Control with Patterns, D-learning, Visual Servoing

## 1 Introduction

In a data-rich age, a system is often in operation when measurements of system inputs and outputs are accessible for collection through inexpensive and numerous information-sensing devices. Based on the input and output data, a direct way is often to model the dynamical system according to the first principles. Then, existing methods are used to analyze the stability

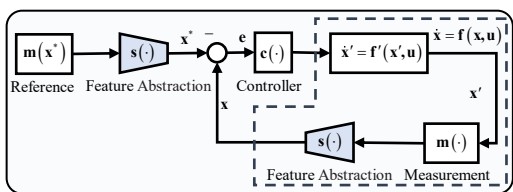

Figure 1: Closed-loop system by CWP.

or design controllers for the identified system. However, there exist two difficulties. First, it is not easy to get the true form of the considered system, so the approximation may not be satisfied. Second, except for only a few experts, the approximated model may still be hard to handle with existing model-based methods.

The development of deep learning and Reinforcement Learning(RL) [1], [2] has led to new advances in these difficulties [3], [4], [5]. The advancement of deep learning and RL has contributed significantly to the development of neural network controllers for robotic systems [6], [7], [8]. For further discussion of related work, please refer to Appendix A.

Despite the impressive performance of these controllers, many of these studies lack critical stability guarantees that are essential for safety-critical applications. To overcome this lack, Lyapunov stability [20] in control theory provides a well-known framework for ensuring the closed-loop stability

---

This paper is a revised version of Quan Quan and Kai-Yuan Cai, Control with Patterns Based on D-learning, https://arxiv.org/abs/2206.03809v2 .

8th Conference on Robot Learning (CoRL 2024), Munich, Germany.

of nonlinear dynamical systems. The core concept of this theory is the Lyapunov function, a scalar function whose value decreases along the closed-loop trajectory of the system. This function represents the process by which the system transitions from the system at any state within the Region of Attraction (ROA) to a stable equilibrium.

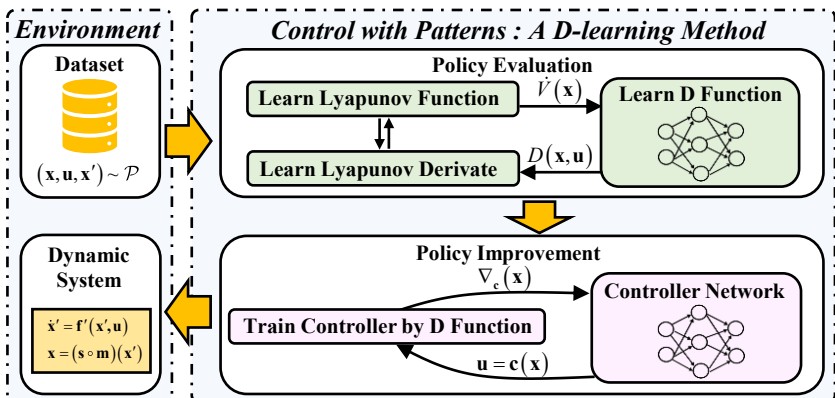

Figure 2: Overview of our method. The main content consists of two parts. For the policy evaluation step, the Lyapunov function and the D-function are updated by solving (13). After learning the D-function, we train the CWP controller by solving (14). For more details, please see Appendix E.

Previous studies [9], [10] that integrate deep learning and Lyapunov control methods have primarily provided guarantees for state feedback control based on structured information (e.g., the state of a linear time-invariant system). Our work addresses the more challenging but practically relevant problem of feedback control based on unstructured information by identifying and overcoming the limitations of existing approaches to synthesize, and certify controllers for real-world applications. In order to demonstrate the effectiveness of our method on real robotic systems, we design a model-free flight controller that can (1) stabilize a hovering multicopter with images as feedback, similar to visual servoing controllers; (2) outperform RL; and (3) provide Lyapunov stability guarantees.

Our key contributions are:

• We propose an approach, namely Control with Patterns (CWP), to the stability problem of dynamical systems, which transforms the controller design problem into a pattern classification problem. CWP represents a novel framework related to Lyapunov function learning, that can be used to develop model-free controllers for general dynamical systems.

• We propose D-learning, which parallels to Q-learning [11] in RL to obtain both Lyapunov function and its derivative (see in Fig.6). Unlike existing Lyapunov function learning methods that rely on controlled models or their approximation with neural networks [12], [13], the system dynamics are encoded in the so-called D-function depending on the actions. This allows CWP to be performed without any knowledge of the system dynamics.

• Results from the simulation platform and real flight experiments show that our approach can stabilize a multicopter with real-time images as feedback. Furthermore, the D-learning trained controller shows superior performance to the RL trained controller.

## 2 Problem Formulation

Consider the following autonomous system

$$\dot{\mathbf{x}}' = \mathbf{f}'\left(\mathbf{x}', \mathbf{u}\right) = \mathbf{f}'\left(\mathbf{x}', \mathbf{c}\left(\mathbf{x}\right)\right)$$
$$\mathbf{x} = \mathbf{s}\left(\mathbf{m}\left(\mathbf{x}'\right)\right)$$

where $\mathbf{x}' \in \mathcal{D}' \subseteq \mathbb{R}^{n'}$ is the original state not available for measurement, $\mathbf{m}\left(\mathbf{x}'\right) \in \mathcal{M}$ is a measurement in the form of unstructured data such as images, $\mathbf{s}\left(\cdot\right)$ is a *feature* selection function

designed to encode the measurement into a vector $\mathbf{x} \in \mathcal{D} \subseteq \mathbb{R}^n$, and $\mathbf{u} = \mathbf{c}(\mathbf{x})$ is the control policy. We want to focus on $\mathbf{x}$ rather than $\mathbf{x}'$, namely, consider the following autonomous system

$$\dot{\mathbf{x}} = \frac{\partial \mathbf{s}}{\partial \mathbf{m}} \frac{\partial \mathbf{m}}{\partial \mathbf{x}'} \mathbf{f}'(\mathbf{x}', \mathbf{u}) \triangleq \mathbf{f}(\mathbf{x}, \mathbf{u}) \tag{1}$$

where $\mathbf{f} : \mathcal{D} \to \mathbb{R}^n, \mathbf{x}(0) = \mathbf{x}_0 \in \mathcal{D}$. From observing the system (1), we can obtain the data set

$$\mathcal{P}_u = \{(\dot{\mathbf{x}}_i, \mathbf{x}_i), i = 1, \cdots, N\}. \tag{2}$$

We prepare to solve the *exponential attraction* (see Definition B.2) problem using the Lyapunov method. The Lyapunov function for the data set (2) is supposed to have the following form

$$V(\mathbf{x}) = \mathbf{g}(\mathbf{x})^{\mathrm{T}} \boldsymbol{\theta}_g \tag{3}$$

where $V(\mathbf{0}_{n \times 1}) = 0$, $V : \mathcal{D}/\{\mathbf{0}_{n \times 1}\} \to \mathbb{R}_+$ and $\boldsymbol{\theta}_g \in \mathcal{S}_g \subseteq \mathbb{R}^{l_1}$. The set $\mathcal{S}_g$ is used to guarantee that the function $V(\mathbf{x})$ is a Lyapunov function. The derivative of $V(\mathbf{x})$ yields

$$\dot{V}(\mathbf{x}) = \left( \frac{\partial \mathbf{g}(\mathbf{x})}{\partial \mathbf{x}} \dot{\mathbf{x}} \right)^{\mathrm{T}} \boldsymbol{\theta}_g. \tag{4}$$

We hope that the derivative satisfies

$$\dot{V}(\mathbf{x}) \leq -W(\mathbf{x}) \tag{5}$$

where $W(\mathbf{x})$ is also a Lyapunov function, which can be further written as

$$W(\mathbf{x}) = \mathbf{h}(\mathbf{x})^{\mathrm{T}} \boldsymbol{\theta}_h$$

where $\boldsymbol{\theta}_h \in \mathcal{S}_h \subseteq \mathbb{R}^{l_2}$. Similarly, the set $\mathcal{S}_h$ is used to guarantee that the function $W(\mathbf{x})$ is a Lyapunov function as well.

For the data set (2), suppose that we have

$$\left( \left. \frac{\partial \mathbf{g}(\mathbf{x})}{\partial \mathbf{x}} \right|_{\mathbf{x}=\mathbf{x}_i} \dot{\mathbf{x}}_i \right)^{\mathrm{T}} \boldsymbol{\theta}_g \leq -\mathbf{h}(\mathbf{x}_i)^{\mathrm{T}} \boldsymbol{\theta}_h \tag{6}$$

where $i = 1, \cdots, N$. Then, in the following, based on (6), we show that the equilibrium state $\mathbf{x} = \mathbf{0}_{n \times 1}$ is *exponentially attractive* on the data set $\mathcal{P}$ in Theorem 2.1. For proof, please see Appendix C.2.

**Theorem 2.1.** *Under Assumptions C.1-C.6, for the system (1), if there exist parameter vectors $\boldsymbol{\theta}_g \in \mathcal{S}_g$ and $\boldsymbol{\theta}_h \in \mathcal{S}_h$ such that (6) holds for the data set $\mathcal{P}$, then the equilibrium state $\mathbf{x} = \mathbf{0}_{n \times 1}$ is exponentially attractive on the data set $\mathcal{P}$.*

Consequently, according to Theorem 2.1, the exponential attraction problem is converted to make the inequality (6) hold. The inequality (6) is rewritten as

$$\mathbf{y}_i^{\mathrm{T}} \boldsymbol{\theta} \geq 0, i = 1, \cdots, N \tag{7}$$

where

$$\mathbf{y}_i = -\left[ \left( \left. \frac{\partial \mathbf{g}(\mathbf{x})}{\partial \mathbf{x}} \right|_{\mathbf{x}=\mathbf{x}_i} \dot{\mathbf{x}}_i \right)^{\mathrm{T}} \quad \mathbf{h}(\mathbf{x}_i)^{\mathrm{T}} \right]^{\mathrm{T}}$$

$$\boldsymbol{\theta} = \left[ \begin{array}{c} \boldsymbol{\theta}_g \\ \boldsymbol{\theta}_h \end{array} \right] \in \mathcal{S} \triangleq \mathcal{S}_g \times \mathcal{S}_h.$$

Formally, according to Theorem 2.1, we construct the CWP problem formulation represented as follows

$$\textit{Design } \mathbf{u} \in \mathcal{U} \textit{ and find } \boldsymbol{\theta} \in \mathcal{S} \textit{ to make (7) hold on the data set (2)} \tag{8}$$

This problem can be classified as a *pattern classification* [14] problem. Here, $\mathbf{y}_i$ is the compound *features* which describe the stability *pattern* for the system (1). Consequently, $f(\boldsymbol{\theta}) = \mathbf{y}^{\mathrm{T}} \boldsymbol{\theta}$ can be regarded as a *linear discriminant function* [14]. So far, we turned the stability problem (6) into the pattern classification problem (8).

# 3 Control with Patterns based on D-learning

After formulating the CWP problem (8), we are going to consider how to construct the model-free controller based on data sets. To this end, we will design a CWP controller based on a proposed D-learning method.

## 3.1 Control with Patterns

In order to solve the CWP problem (8), we solve the following optimization

$$\min_{\eta, \boldsymbol{\theta}_g \in \mathcal{S}_g, a > 0, \mathbf{c}} \quad wa - \eta$$
$$\text{s.t.} \quad -\left( \left. \frac{\partial \mathbf{g}(\mathbf{x})}{\partial \mathbf{x}} \right|_{\mathbf{x}=\mathbf{x}_i} \dot{\mathbf{x}}_i(\mathbf{c}) \right)^{\mathrm{T}} \boldsymbol{\theta}_g - W(\boldsymbol{\theta}_h, \mathbf{x}_i) \geq 0 \qquad (9)$$
$$F_g(\boldsymbol{\theta}_g) \leq a$$

where $i = 1, \cdots, N$, $F_g(\cdot)$ is a constraint on $\boldsymbol{\theta}_g$, $W(\boldsymbol{\theta}_h, \mathbf{x})$ is a Lyapunov function mentioned in (5). In the following for simplicity, let $W(\boldsymbol{\theta}_h, \mathbf{x}) = \eta \|\mathbf{x}\|^2$. An iterative procedure for solving the inequality (9) may be used, including *policy evaluation* and *policy improvement*.

• **Initialization**. Select any admissible (i.e., stabilizing) control $\mathbf{c}_0$, $k = 0$.

• **Policy Evaluation Step**. Under $\mathbf{c}_k$, at state $\mathbf{x}_i$, the control is $\mathbf{u} = \mathbf{c}_k(\mathbf{x}_i) \in \mathcal{U}$, resulting in $\dot{\mathbf{x}}_i(\mathbf{c}_k) \in \mathcal{D}$. Determine the solution $\boldsymbol{\theta}_{g,k}, a > 0, \eta_k$ by

$$\min_{\eta, \boldsymbol{\theta}_g \in \mathcal{S}_g, a > 0} \quad wa - \eta$$
$$\text{s.t.} \quad -\left( \left. \frac{\partial \mathbf{g}(\mathbf{x})}{\partial \mathbf{x}} \right|_{\mathbf{x}=\mathbf{x}_i} \dot{\mathbf{x}}_i(\mathbf{c}_k) \right)^{\mathrm{T}} \boldsymbol{\theta}_g - \eta \|\mathbf{x}_i\|^2 \geq 0 \qquad (10)$$
$$F_g(\boldsymbol{\theta}_g) \leq a$$

where $i = 1, \cdots, N_k$.

• **Policy Improvement Step**. Determine an improved policy using

$$\min_{\mathbf{c}, \eta} \quad -\eta$$
$$\text{s.t.} \quad -\left( \left. \frac{\partial \mathbf{g}(\mathbf{x})}{\partial \mathbf{x}} \right|_{\mathbf{x}=\mathbf{x}_i} \dot{\mathbf{x}}_i(\mathbf{c}) \right)^{\mathrm{T}} \boldsymbol{\theta}_{g,k} - \eta \|\mathbf{x}_i\|^2 \geq 0 \qquad (11)$$

where $i = 1, \cdots, N_k$.

By fixing $\mathbf{x}_i$ for every step $k$, the iteration can be terminated after a sufficient number of steps if $wa_k - \eta_k$ and $-\eta_k$ are nearly not changed. This is due to the fact that the iterative procedure is in fact used to solve the optimization (9) with the coordinate descent [15].

## 3.2 Control with Patterns Based on D-learning

Unfortunately, in the *Policy Improvement Step* (11) , one requires knowledge of the system dynamics $\dot{\mathbf{x}}_i(\mathbf{c})$. To circumvent the necessity of understanding the system dynamics, similar to Q-learning in the field of RL, we can rewrite $\dot{V}(\mathbf{x})$ in (4) as

$$D(\mathbf{x}, \mathbf{u}) = \left( \frac{\partial \mathbf{g}(\mathbf{x})}{\partial \mathbf{x}} \mathbf{f}(\mathbf{x}, \mathbf{u}) \right)^{\mathrm{T}} \boldsymbol{\theta}_g$$

where (1) is utilized. We call it the D-function as it is the *derivative* of the Lyapunov function and it is expected to be *decreased*. If one obtains $D(\mathbf{x}, \mathbf{u})$ by learning directly, then the use of the input coupling function is avoided. In the nonlinear case, it is assumed that the value of $D(\mathbf{x}, \mathbf{u})$ is sufficiently smooth. Referring to [35], according to the Weierstrass higher-order approximation theorem , there exists a dense basis set $\{\varphi_i(\mathbf{x}, \mathbf{u})\}$ such that

$$D(\mathbf{x}, \mathbf{u}) = \sum_{i=1}^{\infty} \theta_i \varphi_i(\mathbf{x}, \mathbf{u}) = \sum_{i=1}^{L} \theta_i \varphi_i(\mathbf{x}, \mathbf{u}) + \sum_{i=L+1}^{\infty} \theta_i \varphi_i(\mathbf{x}, \mathbf{u}) \triangleq \boldsymbol{\theta}_d^{\mathrm{T}} \boldsymbol{\phi}(\mathbf{x}, \mathbf{u}) + \varepsilon_L(\mathbf{x}, \mathbf{u})$$

where basis vector $\boldsymbol{\theta}_d = [\theta_1 \; \theta_2 \; \cdots \; \theta_L]^{\mathrm{T}}$, $\boldsymbol{\phi}(\mathbf{x}, \mathbf{u}) = [\varphi_1(\mathbf{x}, \mathbf{u}) \; \varphi_2(\mathbf{x}, \mathbf{u}) \; \cdots \; \varphi_L(\mathbf{x}, \mathbf{u})]^{\mathrm{T}}$ and $\varepsilon_L$ converges uniformly to zero as the number of terms retained $L \to \infty$.

It is expected to make $D(\mathbf{x}, \mathbf{u}) - \dot{V}(\mathbf{x})$ as small as possible. From a mathematical standpoint, it is imperative to minimize the value of $b$ while adhering to the following constraint

$$\left| \boldsymbol{\theta}_d^{\mathrm{T}} \boldsymbol{\phi}(\mathbf{x}_i, \mathbf{c}_k(\mathbf{x}_i)) - \boldsymbol{\theta}_g^{\mathrm{T}} \left. \frac{\partial \mathbf{g}(\mathbf{x})}{\partial \mathbf{x}} \right|_{\mathbf{x} = \mathbf{x}_i} \dot{\mathbf{x}}_i(\mathbf{c}_k(\mathbf{x}_i)) \right| \leq b$$

where $i = 1, \cdots, N_k$, $k = 1, \cdots, M$.

On the other hand, (5) is rewritten as

$$\boldsymbol{\theta}_d^{\mathrm{T}} \boldsymbol{\phi}(\mathbf{x}, \mathbf{u}) \leq -\eta \|\mathbf{x}\|^2.$$

Furthermore, the inequality (7) is rewritten as

$$\mathbf{y}_i'^{\mathrm{T}} \boldsymbol{\theta}' \geq 0, i = 1, \cdots, N \tag{12}$$

where

$$\mathbf{y}_i' = - \begin{bmatrix} \boldsymbol{\phi}(\mathbf{x}_i, \mathbf{u}_i)^{\mathrm{T}} & \mathbf{h}(\mathbf{x}_i)^{\mathrm{T}} \end{bmatrix}^{\mathrm{T}}$$

$$\boldsymbol{\theta}' = \begin{bmatrix} \boldsymbol{\theta}_d \\ \boldsymbol{\theta}_h \end{bmatrix} \in \mathcal{S}' \triangleq \mathbb{R}^L \times \mathcal{S}_h.$$

With the D function, iterative procedures for solving the inequality (9) should be rewritten, including *policy evaluation* and *policy improvement*.

• **Initialization**. Select any admissible (i.e., stabilizing) control $\mathbf{c}_0$, $k = 0$.

• **Policy Evaluation Step (Based on D-learning)**. Under $\mathbf{c}_k$, at state $\mathbf{x}_i$, the control is $\mathbf{u} = \mathbf{c}_k(\mathbf{x}_i) \in \mathcal{U}$, resulting in $\dot{\mathbf{x}}_i(\mathbf{c}_k) \in \mathcal{D}$. Determine the solution $\boldsymbol{\theta}_{g,k}, \boldsymbol{\theta}_{d,k}, a_k, b_k > 0, \eta_k \in \mathbb{R}$ by

$$\begin{aligned}
&\min_{\boldsymbol{\theta}_d \in \mathbb{R}^L, \boldsymbol{\theta}_g \in \mathcal{S}_g, a, b > 0, \eta \in \mathbb{R}} \quad -\eta + w_1 a + w_2 b \\
&\text{s.t.} \quad -\boldsymbol{\theta}_d^{\mathrm{T}} \boldsymbol{\phi}(\mathbf{x}_i, \mathbf{c}_k(\mathbf{x}_i)) - \eta \|\mathbf{x}_i\|^2 \geq 0 \\
&\qquad \left| \boldsymbol{\theta}_d^{\mathrm{T}} \boldsymbol{\phi}(\mathbf{x}_i, \mathbf{c}_j(\mathbf{x}_i)) - \boldsymbol{\theta}_g^{\mathrm{T}} \left. \frac{\partial \mathbf{g}(\mathbf{x})}{\partial \mathbf{x}} \right|_{\mathbf{x} = \mathbf{x}_i} \dot{\mathbf{x}}_i(\mathbf{c}_j(\mathbf{x}_i)) \right| \leq b \\
&\qquad F_g(\boldsymbol{\theta}_g) \leq a
\end{aligned} \tag{13}$$

where $w_1, w_2 > 0$ are weights, $\dot{\mathbf{x}}_i(\mathbf{c}_j(\mathbf{x}_i)) = \mathbf{f}(\mathbf{x}_i, \mathbf{c}(\mathbf{x}_i))$, $j = 0, \cdots, k, i = 1, \cdots, N_k$.

• **Policy Improvement Step (Based on D-learning)**. Determine an improved policy using

$$\begin{aligned}
&\min_{\mathbf{c}, \eta \in \mathbb{R}} \quad -\eta \\
&\text{s.t.} \quad -\boldsymbol{\theta}_{d,k}^{\mathrm{T}} \boldsymbol{\phi}(\mathbf{x}_i, \mathbf{c}(\mathbf{x}_i)) - \eta \|\mathbf{x}_i\|^2 \geq 0
\end{aligned} \tag{14}$$

where $i = 1, \cdots, N_k$.

## 4 Simulations and Experiments

In this section, simulations and experiments demonstrate that the CWP-based controller can stabilize a hovering multicopter with images as feedback, which can be considered as an IBVS [16] problem.

### 4.1 Simulations Design

#### 4.1.1 Problem Formulation of Visual Servoing

Since the camera is fixed to the body of the multicopter, based on Semi-Autonomous Autopilots (SAAs), the plant [17] can be modeled as

$$\begin{aligned}
\dot{\mathbf{r}} &= \mathbf{v} \\
\mathbf{I} &= \mathbf{m}_{\mathrm{c}}(\mathbf{r})
\end{aligned} \tag{15}$$

where $\mathbf{r}, \mathbf{v} \in \mathbb{R}^3$ indicate the position and velocity of the multicopter, and $\mathbf{I} = \mathbf{m}_{\mathrm{c}}(\mathbf{r})$ represents the mapping from the position $\mathbf{r}$ of the multicopter to the image $\mathbf{I} \in \mathcal{I}$ taken by the camera.

For IBVS, visual servoing can be conceptualized as a minimization problem between the image features $\mathbf{s}(\mathbf{r})$ extracted from the current position $\mathbf{r}$ and the desired features $\mathbf{s}^*$ from the desired position $\mathbf{r}^*$. Then, the controller is decided by

$$\mathbf{e} = \mathbf{s}(\mathbf{I}) - \mathbf{s}(\mathbf{I}^*)$$
$$\mathbf{v} = \mathbf{c}(\mathbf{e}) \tag{16}$$

where $\mathbf{s}(\cdot) \in \mathcal{S}$ is a designed feature selection function, such as neural networks, to code the measurement to a vector in a latent space $\mathcal{S}$; $\mathbf{v} = \mathbf{c}(\mathbf{e})$ represents the controller based on CWP, and its input $\mathbf{e}$ is the feature error between the current image $\mathbf{I} \in \mathcal{I}$ and the desired image $\mathbf{I}^*$.

To extract the features from the current image $\mathbf{I}$ , we use deep metric learning methods, suggested by the work [18], to train the feature selection function. More details about feature extraction can be obtained in Appendix D.

### 4.1.2 D-learning Controller Design

We train the D-learning controller based on the latent space $\mathcal{S}$. The Lyapunov function is designed as a quadratic function, as follows:

$$V(\mathbf{e}) = \mathbf{e}^{\mathrm{T}} \mathbf{P} \mathbf{e} = \mathbf{g}^{\mathrm{T}}(\mathbf{e}) \boldsymbol{\theta}_g \tag{17}$$

where $\mathbf{e} = \mathbf{s}(\mathbf{I}) - \mathbf{s}(\mathbf{I}^*)$, $\mathbf{g}(\mathbf{e}) = \left[\mathbf{e}^{\mathrm{T}} \otimes \mathbf{e}\right]^{\mathrm{T}}$ ($\otimes$ represents Kronecker product), and $\boldsymbol{\theta}_g = \mathrm{vec}(\mathbf{P})$ ,which represents the vectorization of the positive definite matrix $\mathbf{P}$.

Using the Lyapunov function (17), the CWP controller $\mathbf{u} = \mathbf{c}(\mathbf{e})$ is given in Algorithm 1, in which the D-function $D(\mathbf{e}, \mathbf{u})$ and the CWP controller $\mathbf{u} = \mathbf{c}(\mathbf{e})$ is both designed as a 4-layer perceptron, with ReLU activations after each hidden layer.

---

**Algorithm 1:** Control with Patterns Based on D-learning with Constraints

---

**Input:** The data set generated by any admissible (i.e., stabilizing) control $\mathbf{u} = \mathbf{c}_0(\mathbf{e})$
**Output:** CWP controller $\mathbf{u} = \mathbf{c}_k(\mathbf{e})$

1 Initialization. Select any admissible (i.e., stabilizing) control policy parameters $\mathbf{u} = \mathbf{c}_0(\mathbf{e})$ , D-function parameters $\boldsymbol{\theta}_{d,0}$ , Lyapunov function parameters $\boldsymbol{\theta}_{g,0}$, and data set $\mathcal{P} = \{(\mathbf{e}_i, \mathbf{u}_i, t_i), i = 1, \ldots, N\}$ ;

2 **while** *stopping criterion not satisfied $\eta > 0$* **do**

3      Calculate the parameter $\boldsymbol{\theta}_g$ of the Lyapunov function by solving optimization problem

$$\min_{\boldsymbol{\theta}_g \in \mathcal{S}_g, a > 0} \quad wa - \eta$$
$$\text{s.t.} \quad \begin{aligned} &-\left(\mathbf{g}(\mathbf{e}_{i+1})^{\mathrm{T}} \boldsymbol{\theta}_g - \mathbf{g}(\mathbf{e}_i)^{\mathrm{T}} \boldsymbol{\theta}_g\right) - \eta(t_{i+1} - t_i)\|\mathbf{e}_i\|^2 \geq 0 \\ &F_g(\boldsymbol{\theta}_g) \leq a \end{aligned}$$

     where $i = 1, \ldots, N-1$, and $F_g(\cdot)$ represents the constraints on the variable $\boldsymbol{\theta}_g$ ;

4      Estimate the Lyapunov derivative function $\dot{V}(\mathbf{e}_i) = V(\mathbf{e}_{i+1}) - V(\mathbf{e}_i)/(t_{i+1} - t_i)$ ;

5      Update D-function by (13), where $w_1, w_2 > 0$ are weights, $i = 1, \cdots, N-1$;

6      Determine an improved policy $\mathbf{u} = \mathbf{c}_k(\mathbf{e})$ by solving optimization problem (14);

7 **end**

---

## 4.2 Simulations Results

To verify the effectiveness of our control algorithm in the IBVS task, we first perform experimental validation in a simulation environment constructed using RflySim[1].

We sample the camera position in a dimension of $22\,\mathrm{m} \times 8\,\mathrm{m}$ centered on the desired position. Then, the multicopter departs from a set start position and arrives at the desired position by the Position-Based Visual Servoing (PBVS) [16] method based on the position information. By sampling with

---

[1]https://rflysim.com/

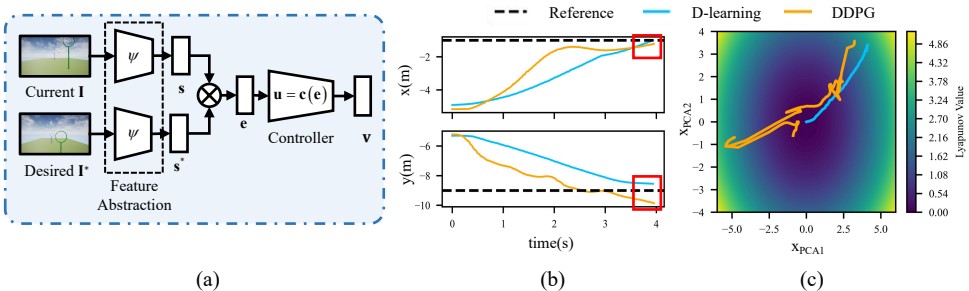

Figure 3: Simulation experiments and results. (a) The multicopter control system. Given the desired image $\mathbf{I}^*$ and the current image $\mathbf{I}$ captured by the camera, the feature error $\mathbf{s}(\mathbf{I}) \in \mathcal{S} \subseteq \mathbb{R}^{32}$ is solved by the neural network $\psi : \mathcal{I} \rightarrow \mathbb{R}^{32}$. The encoding error $\mathbf{e} = \mathbf{s}(\mathbf{I}) - \mathbf{s}(\mathbf{I}^*)$ is used as an input to the controller $\mathbf{u} = \mathbf{c}(\mathbf{e})$. The output of the controller is the velocity $\mathbf{v} \in \mathbb{R}^3$. (b) Performance evaluation of the servo error $\Delta \mathbf{r} = \mathbf{r} - \mathbf{r}^*$ on the simulation environment compared between the controller trained by D-learning and DDPG. The red square shows that the D-learning controller has a smaller error than the DDPG controller. (c) Principal Component Analysis (PCA) projection of the Lyapunov function (17) learned for the system (15), overlaid with the trajectories of the system controlled by the D-learning controller and the DDPG controller, which shows that the D-learning controller has better stability guarantees than the DDPG controller.

equal spacing, 301 trajectories are captured. Based on the collected data tuple $(\mathbf{r}, \mathbf{u}, \mathbf{I}) \in \mathbb{R}^3 \times \mathbb{R}^3 \times \mathcal{I}$, we first train the network $\psi : \mathcal{I} \rightarrow \mathcal{S}$ to obtain $\mathbf{s}(\mathbf{I}) \in \mathbb{R}^{32}$. Then, we use Algorithm 1 to train the controller based on D-learning. Finally, we replace the PBVS controller with the D-learning controller, the experimental results are shown in Fig.3(a). The CWP controller can stabilize the multicopter using images as feedback.

As a comparison, we also train the RL controller. We consider the collected trajectory data as a replay buffer, and use the Deep Deterministic Policy Gradient (DDPG) [19] algorithm to train an actor as a controller. The comparison between the D-learning controller and the DDPG controller is shown in Fig.3(b), in which the DDPG controller, although it can also converge to the reference point, the error is larger than that of the D-learning controller and the Lyapunov function fails to achieve sustained convergence. This result manifests that the D-learning controller provides more reliable stability guarantees than the RL controller.

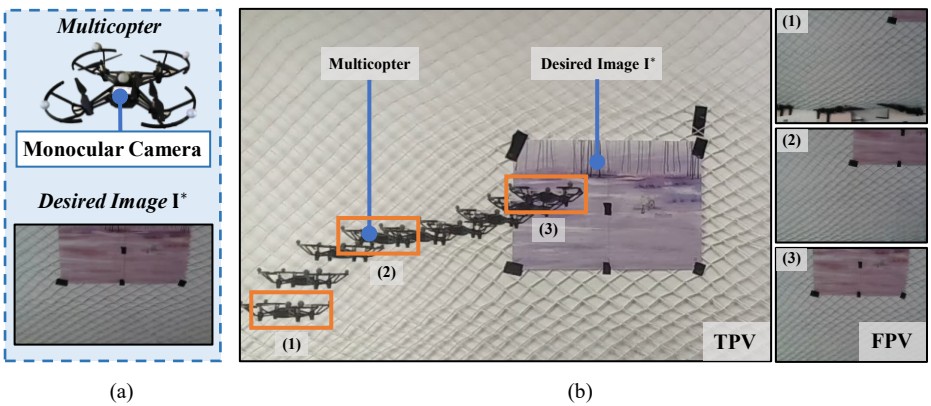

Figure 4: The real flight experiments and results. (a) On the real flight experiments, we use a DJI Tello EDU, which has a front-facing camera. The desired image $\mathbf{I}^*$ of IBVS is centered on a drawing. (b) On the left is the Third-Person-View (TPV) and on the right the First-Person-View (FPV) of the on-board camera. The multicopter positions tend in the order as $(1) \rightarrow (2) \rightarrow (3)$, and the image from FPV converges to the desired image $\mathbf{I}^*$.

### 4.3 Real Flight Experiment

We also deploy our method on a multicopter which features a front-facing camera. We sample the camera position in a dimension of $1.6\,\text{m} \times 0.8\,\text{m}$ centered on the desired position. The target image for image servoing is a painting. The details of real flight experiments are shown in Fig.4. By sampling with equal spacing, 97 trajectories are captured. Based on the collected data tuple $(\mathbf{r}, \mathbf{u}, \mathbf{I}) \in \mathbb{R}^3 \times \mathbb{R}^3 \times \mathcal{I}$, we first train the network $\psi : \mathcal{I} \to \mathcal{S}$ to obtain $\mathbf{s}(\mathbf{I}) \in \mathbb{R}^{32}$. Then, we use Algorithm 1 to train to get the controller based on D-learning. Finally, We replace the controller using position as feedback with a controller using only images as feedback. Experimental results are shown in Fig.5(a). The initial displacement $\Delta\mathbf{r}_0$ is $(-0.60\,\text{m},\ -0.02\,\text{m}, -0.29\,\text{m})$. The D-learning controller can stabilize the multicopter using only images as feedback. 3D trajectory pairs based on the PBVS controller using position as feedback and the D-learning controller using only images as feedback are shown in Fig.5(b).

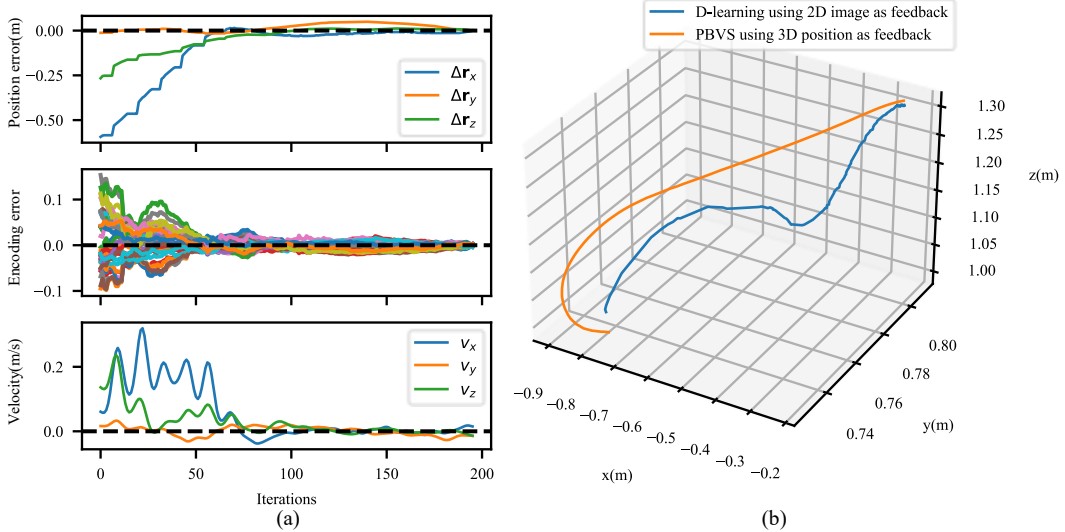

Figure 5: The real flight result for visual servoing. (a) The position error $\Delta\mathbf{r} = \mathbf{r} - \mathbf{r}^*$, the encoding error in the latent space $\mathbf{e} = \mathbf{s}(\mathbf{I}) - \mathbf{s}(\mathbf{I}^*)$, and the velocity of multicopter $\mathbf{v}$ shows the effectiveness of CWP based on D-learning. (b) 3D trajectory based on the PBVS controller using 3D positions as feedback and the D-learning controller using 2D images as feedback.

## 5 Discussion

**Conclusion.** We propose a sampling-based stability condition, *exponential attraction*, to satisfy the Lyapunov stability for learning-based controller. Based on the Theorem 2.1, we propose CWP, which transforms the controller design problem into a pattern classification problem. Subsequently, we propose D-learning for performing CWP in the absence of knowledge regarding the system dynamics. Finally, on both the simulation platform and the real multicopter platform, we show that our approach can synthesize and verify neural network controllers for a control system with images as feedback, and CWP controller has better performance than the controller trained by RL.

**Limitation.** Despite the success in simulated and real flight tasks, our method has not been evaluated in complex practical scenarios. It is anticipated that our approach will be applicable to more complex real-world robotic control tasks, such as locomotion and navigation. To achieve this objective, future work will need to make further improvements in data utilization and provide stability guarantees. Our future work needs to enhance feature extraction in our work, with a particular focus on improving the robustness and generalizability of the feature extraction process. Moreover, the feature function, Lyapunov function, and controller, all in the form of neural networks, can be learned jointly to achieve superior performance. These will be explored in the future work.

**Acknowledgments**

This work is supported by the National Natural Science Foundation of China under Grant 61973015.

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

# A   Related Work

RL [1], [2] and Lyapunov function learning (or certificate learning further, including barrier function and contraction metrics learning) [3], [4], have the potential to handle control problems of complicated systems with big data. Q-learning is a RL method. D-learning, which parallels to Q-learning in RL aims to obtain both Lyapunov function and its derivative. The similarities and differences between the two are illustrated in Fig.6.

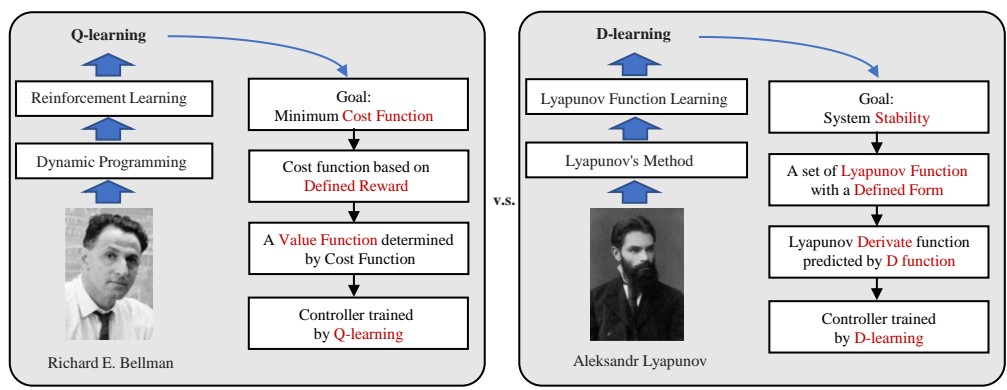

Figure 6: The similarities and differences between Q-learning and D-learning. Q-learning is a RL method, while D-learning is a Lyapunov function learning method.

## A.1   Reinforcement Learning and Lyapunov Function Learning

As a solution to optimal control problems forward-in-time, RL often focuses on optimization based on the Bellman equation [21]. In the traditional control field, the Bellman equation is often used as an analysis tool rather than a direct design tool in optimization control. Nevertheless, the hand-design Lyapunov method, which relies on pseudo-energy functions, is the most prevalent tool in both the analysis and design of control systems. Its objective is to reduce the pseudo-energy functions over time, thereby causing the state to converge to a fixed point. From a technical standpoint, RL is required to define the rewards function and compute the value function (optimal objectives are defined as priors). In contrast, Lyapunov function learning requires training a parameterized Lyapunov function to match the data set (concrete Lyapunov functions are **NOT** defined as priors). Therefore, they are different in both application and design. RL with Lyapunov functions, where certificates are used to ensure safety or stability, has also been proposed recently [8]. A commonly used certificate is the sum of cost over a limited time horizon as a valid Lyapunov candidate [5]. In comparison to RL, Lyapunov function learning offers greater flexibility in candidate selection.

RL based on the Bellman equation is prevalent in the field of computer science due to its model-free characteristics. More importantly, it can solve very complicated control problems. Compared with RL, Lyapunov's methods' achievements on complicated problems with big data are less. Because of the gap between developments by the Lyapunov's method and the Bellman equation, it is hypothesized that there is an increasing focus on Lyapunov function learning from data. The expectation is to unveil its enormous potential, which is also the major motivation of this paper.

The objective of Lyapunov function learning is to construct Lyapunov functions from data. This process can be approached in two main ways:

• Construct a Control Lyapunov Functions (CLF) [22] in formal methods. Lyapunov-stable neural-network control [12], learning-based robust control Lyapunov barrier function [23], neural Lyapunov control [10], and learning-based robust neuro-control [24] employ neural networks to construct both Lyapunov functions and controller simultaneously. These formal methods, which synthesize and verify controllers and Lyapunov functions together, formulate the Lyapunov certification problem

as a proof that certain functions (the Lyapunov function itself, together with the negation of its time derivative) are always non-negative over a domain.

• Learn a certificate in deep learning methods. Demonstration learning [25], [26], episodic learning [27], and imitation learning [28] aim at only searching for a certificate from given control policy data. Notwithstanding the impressive performance of these controllers, many of these controllers require a sufficiently large amount of data to learn semiglobal stabilization. The data collected from actual robotic systems is expensive, which presents a challenge for these controllers.

## A.2 Pattern Classification

Pattern classification is a fundamental process in various fields, including machine learning, data analysis, and computer vision [14]. The objective of pattern classification is to assign a label to a given input based on its characteristics. This process involves categorizing data into predefined classes or categories, making it essential for applications such as speech recognition, image analysis, and bioinformatics. As for the CWP approach, more pattern classification methods and ideas are applicable:

(i) **Methods**. Through the CWP approach, the process of learning model-free controllers with Lyapunov stability guarantees from data can be framed as a pattern classification problem. A variety of pattern classification methods, including support vector machines [29], random forests [30], and decision trees [31], can be employed for the construction of CWP controllers.

(ii) **Features**. CWP employs pattern classification techniques to extract pertinent features from data and construct data-driven controllers. The expression of unstructured data, such as images, videos, and sounds, is challenging due to the difficulty in defining clear physical variables. Pattern classification enables the extraction of low-dimensional structured features from unstructured, high-dimensional data. As a result, features are abstracted from patterns in the form of a state or a set as feedback [32].

(iii) **Negative samples (or counterexamples)**. To date, a number of existing methodologies have employed counterexamples derived from model-based optimization to transform the construction of the Lyapunov function into a pattern-classified problem [33], [34]. In the novel method, it is anticipated that a greater number of negative samples will be generated from data that adheres to the concept of rewards in RL. To illustrate, consider a scenario in which a drone encounters an obstacle or reaches an incorrect equilibrium point through a sequence of actions and states. In such an instance, the series of actions and states will be assigned a negative weight within the range of $[-1, 0]$. As a result of the aforementioned characteristics, a greater number of pattern classification methods are now applicable.

# B    Preliminary Remarks

## B.1    Exponentially Stability and Exponentially Attraction

In this part, some definitions about stability are given related to the system (1) and the data set (2).

**Definition B.1** (Exponentially Stable). For the system (1), an equilibrium state $\mathbf{x} = \mathbf{0}_{n \times 1}$ is *exponentially stable* if there exist $\alpha, \lambda \in \mathbb{R}_+$ such that $\|\phi(\tau; 0, \mathbf{x}_0)\| \leq \alpha \|\mathbf{x}_0\| e^{-\lambda \tau}$ in some neighborhoods around the origin. Global exponential stability is independent of the initial state $\mathbf{x}_0$.

Here, $\phi(\tau; 0, \mathbf{x}_0)$ represents the solution starting at $\mathbf{x}_0$, $\tau \geq 0$. It should be noted that we can only use the data set (2). So, a new definition related to stability, especially for the data set is proposed in the following.

**Definition B.2** (Exponentially Attractive on $\mathcal{P}$). For the dynamics (1), an equilibrium state $\mathbf{x} = \mathbf{0}_{n \times 1}$ is *exponentially attractive* on the data set $\mathcal{P}$ with $\alpha, \lambda, \varepsilon \in \mathbb{R}_+$ if $\|\phi(\tau; 0, \mathbf{x})\| \leq \alpha \|\mathbf{x}\| e^{-\lambda \tau}$, $\forall \tau \geq 0$, $\forall \mathbf{x} \in \mathcal{B}(\mathbf{x}_i, \varepsilon)$ for any $\mathbf{x}_i \in \mathcal{P}$, where $\mathcal{B}(\mathbf{x}_i, \varepsilon)$ denotes a neighborhood around $\mathbf{x}_i$ with radius $\varepsilon$.

Definition B.2 is to describe the trajectory of the system (1) starting from the state. It is hard or impossible to get the *exponential stability* only based on the data set (2) except for more information on $\mathbf{f}(\mathbf{x})$ obtained further. So, the definition of *exponential attraction* especially for the data set can be served as an intermediate result for classical stability results. For some special systems, we can build the relationship between the *exponential stability* and *exponential attraction*.

**Theorem B.3.** *For the autonomous dynamics $\dot{\mathbf{x}} = \mathbf{A}\mathbf{x}$, suppose i) the equilibrium state $\mathbf{x} = \mathbf{0}_{n \times 1}$ is exponential attractive on the data set $\mathcal{P}$ with $\varepsilon$, ii) as shown in Fig.7, $\exists\, l \in \mathbb{R}_+$, $\mathcal{C}_l \subseteq \cup_i \mathcal{B}(\mathbf{x}_i, \varepsilon)$, $\mathbf{x}_i \in \mathcal{P}$, where $\mathcal{C}_l = \left\{ \mathbf{x} \in \mathcal{D} \mid \mathbf{x}^{\mathrm{T}}\mathbf{P}\mathbf{x} = l \right\}$ for a positive-definite matrix $\mathbf{0} < \mathbf{P} = \mathbf{P}^{\mathrm{T}} \in \mathbb{R}^{n \times n}$. Then the equilibrium state $\mathbf{x} = \mathbf{0}_{n \times 1}$ is globally exponential stability.*

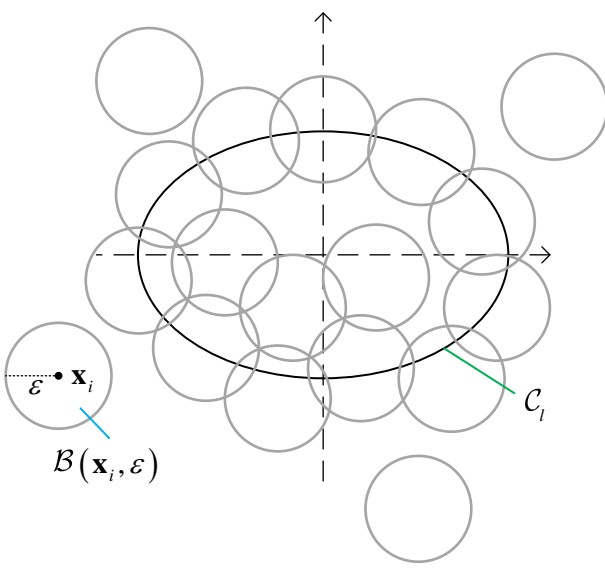

Figure 7: $\mathcal{C}_l$ belongs to the collection of $\mathcal{B}(\mathbf{x}_i, \varepsilon)$.

*Proof.* For any $\mathbf{x}^* \neq \mathbf{0}_{n \times 1} \in \mathbb{R}^n$, since $\mathbf{x}^{*\mathrm{T}}\mathbf{P}\mathbf{x}^* \neq 0$ due to $\mathbf{P}$ being a positive-definite matrix, we have

$$\bar{\mathbf{x}}^* = \theta \mathbf{x}^* \in \mathcal{C}_l$$

where $\theta = \sqrt{\frac{l}{\mathbf{x}^{*\mathrm{T}}\mathbf{P}\mathbf{x}^*}}$. Since $\phi(\tau; 0, \mathbf{x}) = e^{\mathbf{A}\tau}\mathbf{x}$, for $\bar{\mathbf{x}}^* \in \mathcal{C}_l$, the solution is $\phi(\tau; 0, \bar{\mathbf{x}}^*)$ satisfying

$$\phi(\tau; 0, \bar{\mathbf{x}}^*) = \theta \phi(\tau; 0, \mathbf{x}^*).$$

On the other hand, since the equilibrium state $\mathbf{x} = \mathbf{0}_{n \times 1}$ is exponential attractive on the data set $\mathcal{P}$ with $\varepsilon$, there exist $\alpha, \lambda, \varepsilon \in \mathbb{R}_+$ such that $\|\phi(\tau; 0, \bar{\mathbf{x}}^*)\| \leq \alpha \|\bar{\mathbf{x}}^*\| e^{-\lambda\tau}$, $\forall \tau \geq 0$. Therefore,

$$\begin{aligned}
\|\phi(\tau; 0, \mathbf{x}^*)\| &= \frac{1}{\theta} \|\phi(\tau; 0, \bar{\mathbf{x}}^*)\| \\
&\leq \frac{\alpha}{\theta} \|\bar{\mathbf{x}}^*\| e^{-\lambda\tau} \\
&= \alpha \|\mathbf{x}^*\| e^{-\lambda\tau}
\end{aligned}$$

$\forall \tau \geq 0$ for any $\mathbf{x}_i \in \mathcal{P}$. Therefore, the equilibrium state $\mathbf{x} = \mathbf{0}_{n \times 1}$ is globally exponentially stable. $\square$

**Remark 2**. Theorem B.3 implies that, for autonomous linear dynamics, *exponential attraction* is equivalent to *exponential stability* if the data set covers the boundary of an ellipsoid. For general dynamics, the least amount of data required for the equivalence is worth studying. Some research has applied statistical learning theory to provide probabilistic upper bounds on the generalization error, but these bounds tend to be overly cautious [3].

# C Details of Theoretical Analysis

## C.1 Assumptions of Theorem 2.1

**Assumption C.1.** For $\mathbf{x} \in \mathcal{D}$, $\|\mathbf{x}\| \leq d$, where $d \in \mathbb{R}_+$.

**Assumption C.2.** For $\mathbf{x} \in \mathcal{D}$, the function $\mathbf{f}$ satisfies $\|\partial \mathbf{f}(\mathbf{x})/\partial \mathbf{x}\| \leq l_f$, where $l_f \in \mathbb{R}_+$.

**Assumption C.3.** For $\mathbf{x}_1, \mathbf{x}_2 \in \mathcal{D}$, there exists $l_g \in \mathbb{R}_+$ such that

$$\left\| \partial \mathbf{g}(\mathbf{x})/\partial \mathbf{x}|_{\mathbf{x}=\mathbf{x}_1} - \partial \mathbf{g}(\mathbf{x})/\partial \mathbf{x}|_{\mathbf{x}=\mathbf{x}_2} \right\| \leq l_g \|\mathbf{x}_1 - \mathbf{x}_2\|.$$

**Assumption C.4.** For $\mathbf{x} \in \mathcal{D}$, there exist $k_1, k_2 \in \mathbb{R}_+$ such that $k_1 \|\mathbf{x}\|^2 \leq \left\| \mathbf{g}(\mathbf{x})^{\mathrm{T}} \boldsymbol{\theta}_g \right\| \leq k_2 \|\mathbf{x}\|^2$.

**Assumption C.5.** For $\mathbf{x} \in \mathcal{D}$, there exists a $k_3 \in \mathbb{R}_+$ such that $k_3 \|\mathbf{x}\|^2 \leq \mathbf{h}(\mathbf{x})^{\mathrm{T}} \boldsymbol{\theta}_h$.

**Assumption C.6.** $\|\dot{\mathbf{x}}_i\|, \|\mathbf{x}_i\| \neq 0$ for $\forall (\dot{\mathbf{x}}_i, \mathbf{x}_i) \in \mathcal{P}$.

## C.2 Proof of Theorem 2.1

*Proof.* This proof consists of three steps.

*Step 1.* $\|\Delta \dot{\mathbf{x}}_i\| \leq l_f \|\Delta \mathbf{x}_i\|$.

For any $\mathbf{x} \in \mathcal{B}(\mathbf{x}_i, \varepsilon)$, it can be written as

$$\mathbf{x} = \mathbf{x}_i + \Delta \mathbf{x}_i$$

where $\mathbf{x}, \mathbf{x}_i \in \mathcal{D}$ and $\Delta \mathbf{x}_i \in \mathcal{B}(\mathbf{0}, \varepsilon)$. Then $\|\Delta \mathbf{x}_i\| \leq \varepsilon$. In this case, we have

$$\dot{\mathbf{x}} = \dot{\mathbf{x}}_i + \Delta \dot{\mathbf{x}}_i = \mathbf{f}(\mathbf{x}_i + \Delta \mathbf{x}_i) \Rightarrow \Delta \dot{\mathbf{x}}_i = \mathbf{f}(\mathbf{x}_i + \Delta \mathbf{x}_i) - \mathbf{f}(\mathbf{x}_i).$$

Under Assumption C.2, we further have $\|\Delta \dot{\mathbf{x}}_i\| \leq l_f \|\Delta \mathbf{x}_i\|$.

*Step 2.* $\|\phi(\tau; 0, \mathbf{x})\| \leq \alpha \|\mathbf{x}\| e^{-\lambda \tau}$ for $\forall \mathbf{x} \in \mathcal{B}(\mathbf{x}_i, \varepsilon)$.

For any $\mathbf{x} \in \mathcal{B}(\mathbf{x}_i, \varepsilon)$, according to the definition of $V(\mathbf{x})$ in (3), we have

$$
\begin{aligned}
\dot{V}(\mathbf{x}) &= \left( \frac{\partial \mathbf{g}(\mathbf{x})}{\partial \mathbf{x}} \dot{\mathbf{x}} \right)^{\mathrm{T}} \boldsymbol{\theta}_g \\
&= \left( \frac{\partial \mathbf{g}(\mathbf{x})}{\partial \mathbf{x}} \Big|_{\mathbf{x}=\mathbf{x}_i+\Delta \mathbf{x}_i} (\dot{\mathbf{x}}_i + \Delta \dot{\mathbf{x}}_i) \right)^{\mathrm{T}} \boldsymbol{\theta}_g \\
&= \left( \frac{\partial \mathbf{g}(\mathbf{x})}{\partial \mathbf{x}} \Big|_{\mathbf{x}=\mathbf{x}_i} \dot{\mathbf{x}}_i + \frac{\partial \mathbf{g}(\mathbf{x})}{\partial \mathbf{x}} \Big|_{\mathbf{x}=\mathbf{x}_i+\Delta \mathbf{x}_i} (\dot{\mathbf{x}}_i + \Delta \dot{\mathbf{x}}_i) - \frac{\partial \mathbf{g}(\mathbf{x})}{\partial \mathbf{x}} \Big|_{\mathbf{x}=\mathbf{x}_i} \dot{\mathbf{x}}_i \right)^{\mathrm{T}} \boldsymbol{\theta}_g \\
&= \left( \frac{\partial \mathbf{g}(\mathbf{x})}{\partial \mathbf{x}} \Big|_{\mathbf{x}=\mathbf{x}_i} \dot{\mathbf{x}}_i + \frac{\partial \mathbf{g}(\mathbf{x})}{\partial \mathbf{x}} \Big|_{\mathbf{x}=\mathbf{x}_i+\Delta \mathbf{x}_i} \dot{\mathbf{x}}_i - \frac{\partial \mathbf{g}(\mathbf{x})}{\partial \mathbf{x}} \Big|_{\mathbf{x}=\mathbf{x}_i} \dot{\mathbf{x}}_i + \frac{\partial \mathbf{g}(\mathbf{x})}{\partial \mathbf{x}} \Big|_{\mathbf{x}=\mathbf{x}_i+\Delta \mathbf{x}_i} \Delta \dot{\mathbf{x}}_i \right)^{\mathrm{T}} \boldsymbol{\theta}_g \\
&\leq -\mathbf{h}(\mathbf{x}_i)^{\mathrm{T}} \boldsymbol{\theta}_h + l_g \|\Delta \mathbf{x}_i\| \|\dot{\mathbf{x}}_i\| \|\boldsymbol{\theta}_g\| + l_g \|\mathbf{x}_i + \Delta \mathbf{x}_i\| \|\Delta \dot{\mathbf{x}}_i\| \|\boldsymbol{\theta}_g\| \; (\textit{From Assumption C.3}) \\
&\leq -\mathbf{h}(\mathbf{x}_i)^{\mathrm{T}} \boldsymbol{\theta}_h + l_g l_f \|\boldsymbol{\theta}_g\| \|\mathbf{x}_i\| \varepsilon + l_g l_f \|\boldsymbol{\theta}_g\| (\|\mathbf{x}_i\| + \varepsilon) \varepsilon \\
&\leq -\mathbf{h}(\mathbf{x}_i)^{\mathrm{T}} \boldsymbol{\theta}_h + 2 l_g l_f \|\boldsymbol{\theta}_g\| \|\mathbf{x}_i\| \varepsilon + l_g l_f \|\boldsymbol{\theta}_g\| \varepsilon^2 \\
&\leq -k_3 \|\mathbf{x}_i\|^2 + 2 l_g l_f \|\boldsymbol{\theta}_g\| \|\mathbf{x}_i\| \varepsilon + l_g l_f \|\boldsymbol{\theta}_g\| \varepsilon^2. \; (\textit{From Assumption C.5})
\end{aligned}
$$

Then there exists $0 < k_4 < k_3$ and let

$$\varepsilon = -\|\mathbf{x}_i\| + \frac{\|\mathbf{x}_i\| \sqrt{4 l_g^2 l_f^2 \|\boldsymbol{\theta}_g\|^2 + 4(k_3 - k_4) l_g l_f \|\boldsymbol{\theta}_g\|}}{2 l_g l_f \|\boldsymbol{\theta}_g\|} > 0$$

such that $\dot{V}(\mathbf{x}) \leq -k_4 \|\mathbf{x}_i\|^2$. Then, by *Assumption C.1* and *Assumption C.4*, there exists $k_5 = k_4 \|\mathbf{x}_i\| / d^2$ such that

$$\dot{V}(\mathbf{x}) \leq -k_4 \|\mathbf{x}_i\|^2$$
$$\leq -k_5 \|\mathbf{x}\|^2 \ \textit{(From Assumption C.1)}$$
$$\leq -2\lambda V(\mathbf{x}), \forall \mathbf{x} \in \mathcal{B}(\mathbf{x}_i, \varepsilon) \ \textit{(From Assumption C.4)}$$

where $\lambda = k_5 / 2k_2$. Consequently, for $\mathbf{x} \in \mathcal{B}(\mathbf{x}_i, \varepsilon)$, we have

$$V(\phi(\tau; 0, \mathbf{x})) \leq \|V(\mathbf{x})\| e^{-2\lambda\tau}$$

where $\tau \geq 0$.

As a result, by *Assumption C.4*, we have

$$\|\phi(\tau; 0, \mathbf{x})\| \leq \alpha \|\mathbf{x}\| e^{-\lambda\tau}$$

for $\mathbf{x} \in \mathcal{B}(\mathbf{x}_i, \varepsilon)$, where $\alpha = \sqrt{k_2 / k_1}$.

With *Steps 1,2*, there exist $\alpha = \sqrt{k_2 / k_1}$, $0 < \lambda < k_3 / 2k_2$, $\varepsilon > 0$, such that $\|\phi(\tau; 0, \mathbf{x})\| \leq \alpha \|\mathbf{x}\| e^{-\lambda\tau}$, $\forall \tau \geq 0$, $\forall \mathbf{x} \in \mathcal{B}(\mathbf{x}_i, \varepsilon)$. Moreover, $\alpha, \lambda, \varepsilon$ are independent of $\mathbf{x}_i$, so the result is applicable to any $\mathbf{x}_i \in \mathcal{P}$. Therefore, the equilibrium state $\mathbf{x} = \mathbf{0}_{n \times 1}$ is *exponentially attractive* on the data set $\mathcal{P}$.

## D  Details of Feature Extraction

In order to extract the features from the current image $\mathbf{I}$, a ResNet-18 [36] model, trained using metric learning, is employed. The principal objective of metric learning is to develop a novel metric that diminishes the inter-sample distances within a given class and amplifies those between distinct classes.

In order to more accurately represent the feature in question, it is proposed that a multimodal latent space, designated as $\mathcal{S}$, be created in which both position and image representations are mapped. The interrelationship between the latent space, images, and positions is illustrated in Fig.8(a). The position $\mathbf{r}$ maps to a feature embedding $\mathbf{s_r}$, and the image $\mathbf{I}$ acquired at the position $\mathbf{r}$ is noted as $\mathbf{s_I}$.

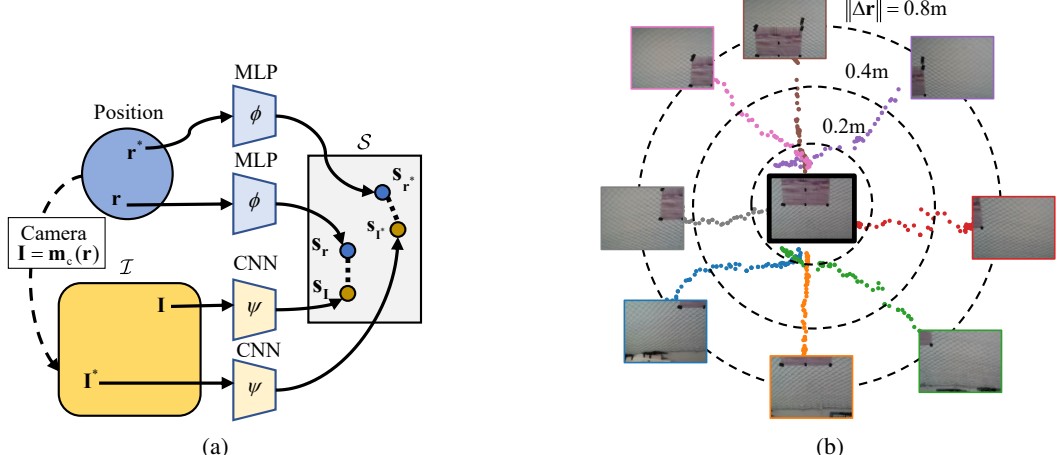

Figure 8: (a) The proposed latent space for visual servoing. Both images and positions are projected in the feature space $\mathcal{S}$, where they can be compared. (b) PCA projection of trajectories $\mathbf{s_I} - \mathbf{s_{I^*}}$ in the latent space, for 2D motions. Circles show the error for the position embeddings $\mathbf{s_r} - \mathbf{s_{r^*}}$ for various distances.

We argue that for the optimal policy of visual servoing, the distance between two embeddings should be equal to the distance between their underlying positions: $d_{\mathcal{S}}(\mathbf{s}_{\mathbf{I}_j}, \mathbf{s}_{\mathbf{I}_j}) = \|\mathbf{r}_j - \mathbf{r}_k\|_2$, where $d_{\mathcal{S}}$ represents the Euclidean distance:

$$d_{\mathcal{S}}(\mathbf{s}_j, \mathbf{s}_k) = \|\mathbf{s}_j - \mathbf{s}_k\|. \tag{18}$$

To learn the space $\mathcal{S}$, we propose to use two distinct, parallel neural networks. The first is $\phi : \mathbb{R}^3 \to \mathcal{S}$, that maps a position $\mathbf{r}$ to an embedding $\mathbf{s}_{\mathbf{r}} = \phi(\mathbf{r})$. The second model $\psi : \mathcal{I} \to \mathcal{S}$, maps an image $\mathbf{I}$ to its latent representation $\mathbf{s}_{\mathbf{I}} = \psi(\mathbf{I})$.

In order to train $\phi$ and $\psi$, we devise our loss function $\mathcal{L}_{\mathcal{S}}$, which is based on the distances between the latent representation of the camera tuple $(\mathbf{r}_j, \mathbf{I}_j)$ and the camera tuple $(\mathbf{r}_k, \mathbf{I}_k)$ by

$$\mathcal{L}_{\mathcal{S}} = \mathcal{L}_{\phi,\mathbb{R}^3} + \mathcal{L}_{\psi,\mathbb{R}^3} + \mathcal{L}_{\phi,\psi} \tag{19}$$

where $\mathcal{L}_{\phi,\mathbb{R}^3}$ is the loss function to train $\phi : \mathbb{R}^3 \to \mathcal{S}$, $\mathcal{L}_{\psi,\mathbb{R}^3}$ is the loss function to train $\psi : \mathcal{I} \to \mathcal{S}$, and $\mathcal{L}_{\phi,\psi}$ is the loss function to shape the feature space $\mathcal{S}$ in the following

$$\mathcal{L}_{\phi,\mathbb{R}^3} = \text{MSELoss}\left(\|\mathbf{r}_j - \mathbf{r}_k\|, \|\mathbf{s}_{\mathbf{r}_j} - \mathbf{s}_{\mathbf{r}_k}\|\right) \tag{20a}$$

$$\mathcal{L}_{\psi,\mathbb{R}^3} = \text{MSELoss}\left(\|\mathbf{s}_{\mathbf{I}_j} - \mathbf{s}_{\mathbf{r}_k}\|, \|\mathbf{s}_{\mathbf{r}_j} - \mathbf{s}_{\mathbf{r}_k}\|\right) \tag{20b}$$

$$\mathcal{L}_{\phi,\psi} = \text{MSELoss}\left(\|\mathbf{s}_{\mathbf{I}_j} - \mathbf{s}_{\mathbf{I}_k}\|, \|\mathbf{s}_{\mathbf{r}_j} - \mathbf{s}_{\mathbf{r}_k}\|\right). \tag{20c}$$

By comparing a representation with each specific tuple, we ensure that a single iteration guides the encoding towards a more stable location. As illustrated in Fig.8(b), the minimization of $\mathbf{e}$ in the latent space results in the emergence of nearly straight lines in the latent space. The error between position embeddings also correlates well with the error from image representations.

In [37], the authors propose the use of autoencoder visual servoing as a method for performing visual servoing in the latent space of an autoencoder. An autoencoder is a neural network architecture comprising an encoder and a decoder. Together, these components learn to project images into low-dimensional representations and then reconstruct them back to the original space. This approach is analogous to that of PCA-based visual servoing [38]. In consideration of two encodings, $\mathbf{s}_{\mathbf{I}}, \mathbf{s}_{\mathbf{I}^*}$, extracted from unstructured data, specifically current image $\mathbf{I}$ and desired image $\mathbf{I}^*$, the neural network-based IBVS control law is expressed as follows:

$$\mathbf{v} = -\lambda \mathbf{L}_{\mathbf{s}_{\mathbf{I}}}^+ (\mathbf{s}_{\mathbf{I}} - \mathbf{s}_{\mathbf{I}^*}) \tag{21}$$

where $\mathbf{L}_{\mathbf{s}_{\mathbf{I}}}$ is computed analytically by applying the chain rule:

$$\mathbf{L}_{\mathbf{s}_{\mathbf{I}}} = \frac{\partial \mathbf{s}_{\mathbf{I}}}{\partial \mathbf{I}} \frac{\partial \mathbf{I}}{\partial \mathbf{r}}$$

and $\mathbf{L}_{\mathbf{s}_{\mathbf{I}}}^+$ represents the Moore-Penrose pseudoinverse of the matrix $\mathbf{L}_{\mathbf{s}_{\mathbf{I}}}$. While this process is typically applied to images, the same rationale can be extended to an encoder for any type of inputs, provided that the interaction matrix of the input can be defined. Neural networks built with PyTorch[2] possess automatic differentiation capabilities, ensuring that the features extracted from images are continuous. This continuity permits their utilization in visual servoing and facilitates the extraction of structured features from unstructured data.

# E  Details of Figure 2

In Fig.2, CWP constructs the model-free controller based on the data set $\mathcal{F}$. Here, $(\mathbf{x}, \mathbf{u}, \mathbf{x}') \sim \mathcal{F}$ represents the collected data tuple of the unstructured data $\mathbf{x}$, the policy $\mathbf{u}$, and the structured feature encoding $\mathbf{x}'$ from the unstructured data $\mathbf{x}$. $V(\mathbf{x})$ represents the Lyapunov function in (3), and $\dot{V}(\mathbf{x}) = \frac{\partial V}{\partial \mathbf{x}} \mathbf{f}(\mathbf{x}, \mathbf{u})$ represents the derivative of Lyapunov function in (4). $D(\mathbf{x}, \mathbf{u})$ represents the D-function in (13). To update the control policy $\mathbf{u} = \mathbf{c}(\mathbf{x})$, CWP computes the gradient $\nabla_{\mathbf{c}}(\mathbf{x})$ to minimize the optimization objective in (14).

---

[2]https://pytorch.org/

# F   Details of Simulations and Real Flight Experiments

Table.1 presents the configuration information of a custom-built multicopter in simulations and Table.2 presents the configuration information of a Tello EDU multicopter[3] in real flight experiments. We use the custom built multicopter for data collection and validation in simulations, and use the Tello EDU multicopter for data collection and validation in the real flight experiments. For communication, we use DJITelloPy[4] as the upper-level flight control communication program.

For data collection, we use a motion capture system to obtain the position $\mathbf{r} \in \mathbb{R}^3$ and velocity $\mathbf{v} \in \mathbb{R}^3$ of the multicopter. For real flight experiments, we set Remote Controller(RC) to control the velocity $\mathbf{v} \in \mathbb{R}^3$ of the multicopter.

To ensure real-time control, we use CUDA[5] to speed up the inference process. In simulations conducted on a platform with an Intel i7-13700K CPU and an RTX 4070 Ti GPU, the neural network inference frequency exceeds $100\,\mathrm{Hz}$, thereby meeting the requisite standards for real-time image feedback control.

|  | custom built multicopter |
| --- | --- |
| Weight | $1.5\,\mathrm{kg}$ |
| Airframe radius | $0.225\,\mathrm{mm}$ |
| Diagonal size | $450\,\mathrm{mm}$ |
| Camera frame rate | 30 Fps |
| Camera resolution | $720 \times 405$ |
| Configuration | X4 |
| Flight controller in simulations | CopterSim running PX4 |

Table 1: Configurations of the custom built multicopter in simulations.

|  | Tello EDU multicopter |
| --- | --- |
| Weight | $0.08\,\mathrm{kg}$ |
| Size | $98\,\mathrm{mm} \times 92.5\,\mathrm{mm} \times 41\,\mathrm{mm}$ |
| Propeller size | 3 inches |
| Camera frame rate | 30 Fps |
| Camera resolution | $960 \times 720$ |

Table 2: Configurations of the Tello EDU multicopter in real flight experiments.

---

[3]https://www.ryzerobotics.com/tello-edu
[4]https://github.com/damiafuentes/DJITelloPy
[5]https://developer.nvidia.com/cuda-toolkit

