# OpenReview forum: "Control with Patterns: A D-learning Method"
_robot-learning.org/CoRL/2024/Conference — CoRL 2024_

### Official Review · Reviewer_6cXy · 2024-07-22
**Good paper**

**Originality:** 4
**Technical Quality:** 3
**Clarity Of Presentation:** 4
**Potential Impact:** 2
**Recommendation:** 3
**Confidence:** 4

**Review:**

The following review is organized as follows: at first, a general opinion on the form of the paper is given; in second place, a discussion of the theory and the related comments are presented; lastly, the simulation and experimental section is discussed.
The presented paper is easy to read, comes with a clear flow and a coherent structure. Some supplementary material is given to the reviewer - although not the codes.
The strength of this paper consists in its theory development and the related novelty. As anticipated in the summery of the paper, the theoretical contributions are mainly twofold: a Control with Patterns - CWP - novel algorithm is proposed, related to Lyapunov function learning, that can be used to develop model-free controllers for general dynamical systems for the stability issue of dynamical systems, which transforms the controller design problem into a pattern classification problem; a D-learning method is proposed, where the system dynamics are encoded into the D function depending on actions, allowing to perform CWP without the knowledge of system dynamics.
The problem is clearly formulated in secrions 2 and 3, and in the following appendices. The theoretical model development is interesting, complete and well formulated, includeing also pseudo-codes and algorithm sections.
On one hand, section number 4, which presents simulations and experiments, includes a good presentation of model formulation, concerning the dynamics of the problems at hand.
On the other hand, the numerical results, simulation and experimental results, the testbed used in the case of real-flight experiments could be explained with more detail. Especially when it comes to real-life applications, it would be worth to showcase all of the experiment setup in much more detail.
The lack of real-life application details unfortunately impacts the "potential impact" score.

**Quality Of The Limitations Section:**

2

**Questions For Rebuttal:**

An extension of the experimental result sections is strongly recommended, in particular: in terms of showcasing and commenting the experimental results - for both simulated and real-flight experiments - and in terms of the describing the testbed used for real flight experiments - and, if possible, motivating the choice of the latter with respect to, for example, a different setup.

**Robotics Focus:**

4

**Summary Of Paper:**

The purpose of the paper is twofold, at first, it proposes a novel method for control, namely Control with Patterns - CWP, to address the stability issues over datasets corresponding to non-linear dynamical systems, to mitigate the challenges of Lyapunov methods employed in learning-based controllers applied to such datasets. For this kind of datasets, a new definition is formulated, namely, "exponential attraction" to describe such non-linear dynamical systems.  In second place, a D-learning algorithm is proposed to perform CWP, when the underlying dynamics knowledge of the system is not given. Experiments are showcased - both in a simulated and a real-world scenario.

**Summary Of Recommendation:**

Given the clarity of the theoretical presentation, the novelty and the originality of the topic, and the interesting - although irriproducibile - experimental section, the reviewer suggests to accept the paper for publication.

---

### Official Review · Reviewer_5T6L · 2024-07-27
**Learning Image conditioned stable controllers with D-Learning**

**Originality:** 3
**Technical Quality:** 3
**Clarity Of Presentation:** 3
**Potential Impact:** 3
**Recommendation:** 3
**Confidence:** 2

**Review:**

**Strenghts**
- The problem is highly relevant for robotics. Learning stable controllers from images is highly important for the field of robot learning as a whole.
- The authors present a mathematically detailed description of the derivations of their method, allowing the reader to understand the selection they made.

**Weaknesses**
- The presented method seems to rely largely on the extracted features from the images as the controller is trained to minimize the distance between desired features and target features. This makes the feature selection an essential part in the performance of the robot. However, the paper gives very few details on the extraction of them. The paper would benefit from an extended description of the features and the neccesary properties of these features for proper performance.
- It is not clear if the stable controller is trained to match a single target image or to follow a given trajectory. I understand that the controller is trained to track a demonstration and the target is used as goal conditioning, but further clarifications would be needed.

**Quality Of The Limitations Section:**

2

**Questions For Rebuttal:**

- What are the properties the features should satisfy? Are they required to be smooth?
- Is the controller guaranteed to perform well out of demonstrations area? If the dynamics are unknown, how is possible to train a model in regions where u lack demonstrations?

**Robotics Focus:**

4

**Summary Of Paper:**

The authors introduce a novel algorithm to learn stable policies via pattern classification. To learn under unknown dynamics, they propose a method based on D-Learning.

**Summary Of Recommendation:**

The paper presents an interesting problem and interesting solution. Nevertheless, there is some missing information regarding the feature selection that authors should explain in more detail.

---

### Official Review · Reviewer_FHr9 · 2024-07-30
**Recommend for acceptance**

**Originality:** 5
**Technical Quality:** 4
**Clarity Of Presentation:** 3
**Potential Impact:** 4
**Recommendation:** 4
**Confidence:** 3

**Review:**

The integration of Lyapunov theory with deep learning presented in this paper is both fascinating and innovative. The authors have successfully combined a robust theoretical foundation with substantial experimental validation. Here are a few comments and suggestions for further improvement:

1. Figure 2 Clarification: In Figure 2, it would be beneficial to include definitions and explanations for the variables in the caption, such as x, u, x', V(x), etc. This will enhance the clarity and understanding of the figure for readers.

2. Pattern Classification Background: Providing a more comprehensive introduction to pattern classification would be appreciated. This additional background will help readers who may not be familiar with this concept to better grasp the significance and application of pattern classification in the context of this research.

3. System Architecture: Including a detailed system architecture diagram would significantly improve the paper. This diagram should illustrate the parameters outputted by the model and how these parameters are used to control the drone effectively. A clear visual representation of the system’s workflow will aid in understanding the overall approach.

4. Technical Details: Incorporating technical details, such as the camera framerates (in Hz) and model computation times (in milliseconds), is crucial. These details provide insight into the practical implementation and performance of the system, which is essential for readers looking to replicate or build upon this work.

5. Real-Time Control: It is important to elaborate on the measures taken to ensure that the model output is fast enough to meet real-time control requirements. This could include techniques for optimizing computation speed or strategies for handling delays. Providing this information will address potential concerns about the system’s responsiveness and reliability.

6. Training Data: Detailed information about the type of data used to train the model should be included. This could cover aspects such as the nature of the data, the diversity of the scenarios covered, and the preprocessing steps taken. Understanding the training data is crucial for evaluating the model’s robustness and generalizability.

In summary, the paper presents a unique and compelling integration of Lyapunov theory with deep learning. By incorporating these additional details and clarifications, the paper would provide a more comprehensive and accessible presentation of the research, ultimately benefiting a wider audience.

**Quality Of The Limitations Section:**

3

**Questions For Rebuttal:**

The integration of Lyapunov theory with deep learning presented in this paper is both fascinating and innovative. The authors have successfully combined a robust theoretical foundation with substantial experimental validation. Here are a few comments and suggestions for further improvement:

1. Figure 2 Clarification: In Figure 2, it would be beneficial to include definitions and explanations for the variables in the caption, such as x, u, x', V(x), etc. This will enhance the clarity and understanding of the figure for readers.

2. Pattern Classification Background: Providing a more comprehensive introduction to pattern classification would be appreciated. This additional background will help readers who may not be familiar with this concept to better grasp the significance and application of pattern classification in the context of this research.

3. System Architecture: Including a detailed system architecture diagram would significantly improve the paper. This diagram should illustrate the parameters outputted by the model and how these parameters are used to control the drone effectively. A clear visual representation of the system’s workflow will aid in understanding the overall approach.

4. Technical Details: Incorporating technical details, such as the camera framerates (in Hz) and model computation times (in milliseconds), is crucial. These details provide insight into the practical implementation and performance of the system, which is essential for readers looking to replicate or build upon this work.

5. Real-Time Control: It is important to elaborate on the measures taken to ensure that the model output is fast enough to meet real-time control requirements. This could include techniques for optimizing computation speed or strategies for handling delays. Providing this information will address potential concerns about the system’s responsiveness and reliability.

6. Training Data: Detailed information about the type of data used to train the model should be included. This could cover aspects such as the nature of the data, the diversity of the scenarios covered, and the preprocessing steps taken. Understanding the training data is crucial for evaluating the model’s robustness and generalizability.

In summary, the paper presents a unique and compelling integration of Lyapunov theory with deep learning. By incorporating these additional details and clarifications, the paper would provide a more comprehensive and accessible presentation of the research, ultimately benefiting a wider audience.

**Robotics Focus:**

4

**Summary Of Paper:**

The paper addresses the challenge of ensuring formal (Lyapunov) stability guarantees for learning-based controllers in nonlinear dynamical systems. It introduces a novel control approach called Control with Patterns (CWP) to tackle this issue. The key contributions are:  Exponential Attraction on Data Sets: A new definition to describe nonlinear dynamical systems is proposed. Pattern Classification: The problem of exponential attraction is converted into a pattern classification problem using parameterized Lyapunov functions. D-Learning Method: This method allows the implementation of CWP without prior knowledge of system dynamics. Empirical Validation: The effectiveness of CWP and D-learning is demonstrated through simulations and real flight experiments, particularly in stabilizing the position of a multicopter using real-time images for feedback, akin to an Image-Based Visual Servoing (IBVS) problem.

**Summary Of Recommendation:**

Recommendation for acceptance, supported by its unique idea and evaluation.

---

### Author Rebuttal · Authors · 2024-08-13

We extend our gratitude to the reviewers for their insightful feedback. Their recognition of the novelty (6cXy, 5T6L and FHr9) and technical solidness (6cXy, 5T6L and FHr9) and impressiveness of our experiments (FHr9), is truly encouraging.

Our revision contains the following major updates:
* Added background of pattern classification. We provided a more comprehensive introduction to pattern classification ,and the features that can be addressed by applying pattern classification methods in control. (In Appendix A.3)
* Added the motivation and technical details for using deep metric learning to extract features from real-time images for image-based visual servoing (IBVS). (In Appendix D)
* Fixed unclear information in Fig. 2, such as $x,u,x',V(x)$ . (In Appendix E)
* Added more technical details on the simulation and real flight experiments, such as camera frame rate, model computation times, and measures taken to ensure real-time control. (In Appendix F)
* Added the relationship diagram between Q-learning and D-learning. (In Figure 6)

We appreciate the reviewers' feedback(Reviewer 6cXy and 5T6L) and have incorporated their suggestions, including those related to feature extraction and additional experiments, into the future work section. (In Section 5)

We have highlighted our major updates in blue and attached the appendix to the main paper.

---

### Decision · Program_Chairs · 2024-09-04

**Decision:**

Accept

**Comment:**

The paper was evaluated positively by the reviewers. Raised concerns were addressed in an updated version of the paper.

Here is a high level overview of the reviews. (before rebuttal)
### Strengths:
- The integration of Lyapunov theory with deep learning is both fascinating and innovative, combining a robust theoretical foundation with substantial experimental validation. (Reviewer FHr9)
- The problem addressed is highly relevant for robotics, particularly learning stable controllers from images, which is crucial for robot learning. (Reviewer 5T6L)
- The paper is well-organized, easy to read, and comes with a clear flow and coherent structure, including a mathematically detailed description of the method. (Reviewer FHr9, Reviewer 6cXy)
- The theoretical contributions are novel and interesting, presenting a new control approach (CWP) and a D-learning method to perform CWP without prior knowledge of system dynamics. (Reviewer FHr9, Reviewer 6cXy)

### Weaknesses:
- The paper lacks detailed explanations for the features extracted from images, which are crucial for the controller's performance. (Reviewer 5T6L)
- There is insufficient detail on the experimental setup, particularly for real-life applications. (Reviewer 6cXy)
- The paper requires additional technical details such as camera framerates, model computation times, and measures taken to ensure real-time control. (Reviewer FHr9)
- The training data and its diversity, as well as the preprocessing steps, are not adequately described, which is important for evaluating the model's robustness and generalizability. (Reviewer FHr9)

The weaknesses have been mostly addressed in the rebuttal.